# Individual Social Capital and Life Satisfaction among Mainland Chinese Adults: Based on the 2016 China Family Panel Studies

**DOI:** 10.3390/ijerph18020441

**Published:** 2021-01-08

**Authors:** Wenjun Li, Haiyan Sun, Wen Xu, Wenyuan Ma, Xin Yuan, Yaling Niu, Changgui Kou

**Affiliations:** 1Department of Social Medicine and Health Management, School of Public Health, Jilin University, No. 1163 Xinmin Street, Changchun 130021, China; sunhy19@mails.jlu.edu.cn (H.S.); xuwen18@mails.jlu.edu.cn (W.X.); wyma20@mails.jlu.edu.cn (W.M.); yuanxin20@mails.jlu.edu.cn (X.Y.); 2Department of Epidemiology and Biostatistics, School of Public Health, Jilin University, No. 1163 Xinmin Street, Changchun 130021, China; niuyl19@mails.jlu.edu.cn (Y.N.); koucg@jlu.edu.cn (C.K.)

**Keywords:** life satisfaction, social capital, general trust, norms of reciprocity, neighborhood relationship, chinese adults, China Family Panel Studies (CFPS)

## Abstract

Background: At present, most Chinese adults are under great psychological pressure, which seriously affects the improvement of life satisfaction. The purpose of this study was to shed light on the associations between individual social capital and life satisfaction among mainland Chinese adults. Methods: We used a nationally representative dataset called 2016 China Family Panel Studies, and a total of 26,009 people suited our study requirements. Chi-squared test and binary regression analysis were used to determine the relationship between individual social capital and life satisfaction among mainland Chinese adults. Results: The results indicated that cognitive social capital had significant effects on life satisfaction (*p* < 0.05), but the impact of structural social capital on life satisfaction became less significant when combined with sociodemographic variables and socioeconomic status (SES) (*p* > 0.05). Moreover, we also found that life satisfaction was better in married/cohabited (*p* < 0.05) and in over 65 age group people (*p* < 0.05). Self-rated income level, self-rated social status, and self-rated health all had positive effects on life satisfaction (*p* < 0.001). Conclusions: Given the stabilizing effect of cognitive social capital at the individual level on life satisfaction, the government should attach great importance to this aspect when trying to improve adults’ life satisfaction.

## 1. Introduction

Life satisfaction is a subjective assessment of marriage, work, life, and other areas based on self-defined criteria, which can be positive or negative [1,2]. As one of the important indicators to measure the quality of life, life satisfaction includes not only physical and mental health, but also social adaptability, which can comprehensively evaluate people’s life. There are various factors that affect adults’ life satisfaction in the modern society, such as social support [3] and occupational stress [4]. Actually, with the acceleration of Chinese economy and urbanization, the proportion of Chinese adults living in unhappiness is increasing [5]. Specifically, the implementation of the Chinese family planning policy has led to the emergence of the only child in families, who shoulder the heavy burden of supporting their parents and raising their children. These conditions will eventually aggravate psychological burden and inhibit the improvement of life satisfaction. Therefore, it is imperative for us to pay attention to the current situation of life satisfaction among mainland Chinese adults.

With the development of social determinants of health, the role of social capital in human quality of life has been increasingly recognized [6]. Many scholars have their own unique views on social capital. Bourdieu [7] defined social capital from the perspective of social networks and pointed out that social capital is a collection of actual or potential resources inseparable from the enduring network of relationships. On the basis of Bourdieu, Coleman [8,9] defined social capital from its specific functions and pointed out that social capital is not a single entity, but rather different entities with various forms. Putnam [10] noted that “social capital” refers to “features of social organization”, in terms of trust, norms of reciprocity, and social network. In addition to the above three different definitions of social capital, other authors also contribute to the enrichment and development of social capital. For example, Granovetter [11] put forward the theory of “weak ties”, in which individuals connect with other social circles to form bridges that can provide valuable information that is not available in their own circles. Lin [12,13] believed that social capital is a resource embedded in social networks and social relationship, which can be acquired or flowed through purposeful actions. To sum up, the definition of social capital has not been unified.

Moreover, social capital is often used to examine its positive effects on mental health in terms of its structural or cognitive components, such as depression [14] and life satisfaction [15,16]. The cognitive dimension refers to the evaluation of people’s perception of trust, reciprocity, and support, which represents how people feel inside; the structural dimension refers more to the interaction with other people such as social network, social relationship, and organizational participation, and so on, which represents what people do [17,18,19] There is a growing consensus that empirical indicators of social capital can be measured by social networks (e.g., informal relationship, volunteering, organizational participation), social norms (e.g., shared norms, civic values), and trust (e.g., institutional trust, interpersonal trust, generalized trust) [20]. Similarly, some evidence can also be found to prove social capital is included in the above definition.

Recently, there have been increasing studies on how social capital affects life satisfaction [21,22]. The existing studies have provided evidence of the relationship between social capital and life satisfaction at the individual and aggregate levels. For example, a study used data from the European Social Survey and analyzed the associations between social capital (both at the individual and aggregate levels) and life satisfaction. In addition to the positive impact of individual social capital (such as trust) on life satisfaction, social capital was positively correlated with life satisfaction at the aggregate level [20]. Another study from 50 countries also found similar results, and pointed out that the benefits of social capital to life satisfaction depended on the country’s social capital [23].

The necessity of studying the relationship between individual social capital and life satisfaction is mainly considered from the following several aspects. First, previous studies in China either selected specific population as the research object [24,25] or limited study area [15,26]. For example, Pang [27] investigated the impact of social capital on life satisfaction among Chinese overseas students in Germany and found that social capital is a significant predictor of life satisfaction. Yuan [28] only analyzed the impact of social capital on life satisfaction in three Chinese cities (Shanghai, Beijing, and Guangdong).

Second, there are different results about the relationship between structural social capital and life satisfaction. A study analyzed the relationship between structural social capital and life satisfaction of 6002 residents in three regions of China, and concluded that structural social capital had a positive effect on life satisfaction [28]. In a survey of rural China, Pan [15] found that one of the dimensions of structured social capital (organizational membership) had a significant negative impact on life satisfaction. However, in another related study of residents in East Asia, structured social capital (organizational membership) did not show a significant relationship with life satisfaction [29]. In short, the impact of structural social capital on life satisfaction remains to be studied.

Last not but least, most of the existing research studies on social capital and life satisfaction were based on the background of developed countries, but few studies were carried out on developing countries. Great changes have taken place in China since the reform and opening up. Understanding the relationship between individual social capital and life satisfaction in mainland Chinese has guiding significance for improving the national happiness index in the future.

Hence, this paper focused on the relationship between individual social capital and mainland Chinese adults’ life satisfaction based on nationally representative data. Social capital, in this study, was measured by organizational participation, general trust, norms of reciprocity, and neighborhood relationship. Among them, organizational participation belonged to structural social capital, and the rest belonged to cognitive social capital. Furthermore, “norms of reciprocity” was composed of reciprocity and unfair treatment. Additionally, variables related to socioeconomic status (SES) were selected for inclusion in our study. In short, we provided through this study reliable evidence for the establishment of domestic relevant social policies aimed at improving adults’ life satisfaction by adjusting their social capital.

## 2. Materials and Methods

### 2.1. Data Source and Sample Composition

The data were taken from China Family Panel Studies (CFPS), funded by the “985 Program” of Peking University, Beijing, China and the National Natural Science Foundation of China. CFPS is a national longitudinal general social survey project maintained by the Institute of Social Science Survey (ISSS) of Peking University. The national baseline survey began in 2010 and has been conducted every two years since. It includes 36,892 Chinese respondents residing in 621 villages or communities from 25 of 30 Chinese provinces. Taking into account regional differences and survey costs, CFPS implemented probability proportional to size (PPS) sampling with implicit stratification [30]. Each subsample in CFPS went through three stages of extraction (districts/counties–villages/communities–households). It used computer-assisted personal interviews (CAPIs) during the survey to improve work efficiency and adopted a telephone check, a field check, an audio record check, interview reviews, and statistical analysis to ensure data quality. In this study, we used the adult questionnaire and its target respondents were adults aged 16 years old or above. The questionnaire collected detailed information about social capital and life satisfaction of mainland Chinese adults. After cleaning the missing values, our final sample size was 26,009.

### 2.2. Life Satisfaction

Consistent with previous studies [22,31,32], life satisfaction was measured by a single question: “In general, how satisfied are you with your present life?”, answered on a 5-point Likert scale ranging from 1 (very dissatisfied) to 5 (very satisfied). We drew lessons from the previous research on the classification of life satisfaction [24,33]. Thus, we further categorized these choices into two groups, namely dissatisfied (answers from 1 to 3) and satisfied (answers of 4 or 5). Meanwhile, previous studies from different cultures and countries have pointed out that such a method was considered a reliable measurement of life satisfaction [24,34,35].

### 2.3. Social Capital

#### 2.3.1. General Trust

In our study, “general trust” was assessed by the following question: “Generally speaking, do you think most people are trustworthy or suspicious?”.

#### 2.3.2. Norms of Reciprocity

“Reciprocity” was assessed by the following question: “Do you think most people are helpful or selfish?”. “Unfair treatment” was based on whether the respondent experienced the following situations, including unfair treatment due to poverty gap, gender, and household register; unfair treatment by government cadres; conflict with government personnel; delay in handling affairs in government departments; unreasonable charges by the government. If the respondent met one of the situations, they were classified as having experienced one.

#### 2.3.3. Neighborhood Relationship

“Neighborhood relationship” was based on the following question: “Overall, how do you feel about your neighborhood?”, answered on a 5-point Likert scale ranging from very poor to very good. Furthermore, we categorized these options into two groups, namely poor (answers: “very poor”, “poor”, and “fair”) and good (answers: “good” and “very good”).

#### 2.3.4. Organizational Participation

“Organizational participation” was based on whether the respondent joined the following organizations, including the Communist Party of China, the Communist Youth League of China, Labor Union, Religious Groups, and Individual Laborers’ Association. If the respondent was a member of any of the above organizations, they were classified as “yes”.

### 2.4. Socioeconomic Status

In previous studies, SES was mostly measured by educational attainment and income [36,37]. Therefore, in addition to years of education and self-rated income level, we added self-rated social status as a measure of SES in this study. Based on the education system in China, we divided years of education into the following four groups: 1: ≤6; 2: ~9; 3: ~12 and 4: ≥13.

### 2.5. Sociodemographic Variables

Sociodemographic variables were constituted by age, gender, registered residence (rural and urban), marital status (never married, married/cohabited, and divorced/widowed), religious faith (no and yes), type of work (agriculture, non-agriculture, and inapplicable), and self-rated health (healthy, general, and unhealthy).

### 2.6. Statistical Analysis

We used IBM SPSS Statistics (version 24.0, Armonk, NY, USA) to process data. A *p* < 0.05 was considered statistically significant and a 95% confidence interval was provided for analysis when appropriate. Since the method of multistage probability sampling was adopted to choose samples, we assigned the sample weight when calculating to ensure the scientific rationality of the data. We initially described the overall distribution of the total sample. Then, chi-squared test was used to carry out univariate analysis. Finally, binary logistic regression was used to assess the associations between individual social capital and life satisfaction.

## 3. Results

Overall, 58.9% of the people are satisfied with their life. Table 1 shows the general description of the sample. Compared with agricultural work (32.8%), there are slightly more people engaged in non-agricultural work (37.2%). More than half of the respondents report that they are in good health (65.8%) and have no religious faith (87.7%). Most of our respondents are married or cohabited (75.0%), come from rural areas (68.0%), and have nine years of education or less (70.2%). Most people think that their income level and social status are at a general level in the local area, accounting for 36.2% and 45.9% of the total, respectively. More than half of the respondents believe that most of the people are trustworthy (57.0%) or helpful (73.0%), and have never experienced unfair treatment (73.4%). Most people maintain good relationships with neighborhoods (63.0%) and more than 65% of people are not members of any of the social organizations we mentioned.

Table 2 shows the distribution of life satisfaction among different characteristic groups. Through univariate analysis, all variables except religious faith and gender proved to have significant impact on adults’ life satisfaction (*p* < 0.05). More specifically, there is a similar U-shaped distribution between age and life satisfaction. People who are currently married or cohabited show the highest percentage of life dissatisfaction (42.1%). People with 7–9 years of education are the most dissatisfied with life (44.9%). Self-rated health, self-rated social status, and self-rated income level all had significant positive effects on life satisfaction. In terms of social capital factors, Table 2 shows that the respondents who think most of the people are trustworthy or helpful, and who have never experienced unfair treatment are more satisfied with their life, and the satisfied rate is 62.5%, 62.4%, and 62.9% respectively. In addition, about 64.6% of people who maintain a friendly relationship with their neighbors think they are satisfied with their life at present. People who are members of social organizations (63.5%) are more satisfied with their life than people who are not.

The results of binary logistic regression analysis are shown in Table 3. First, Model 1 only includes social capital, which reports that all variables are significantly associated with life satisfaction (*p* < 0.05). Then, SES is added into Model 2. The odds ratio of the social capital variables only slightly changes. Each variable of SES has a significant impact on life satisfaction (*p* < 0.001). Eventually, Model 3 introduces sociodemographic variables based on Model 2. With the introduction of sociodemographic variables and SES into the model, the significant effect of organizational participation on life satisfaction gradually weakens and finally becomes no longer significant. The other indicators of social capital remain significant and maintain a stable impact on life satisfaction (*p* < 0.05). In addition, compared with respective reference items, people aged over 65 years old (OR = 1.272; 95% CI = 1.000–1.619) and married or cohabited (OR = 1.364; 95% CI = 1.138–1.635) are more likely to think that their current life is satisfactory. Consistent with the results of univariate analysis, self-rated health, self-rated income level, and self-rated social status have positive effects on life satisfaction (*p* < 0.001). Moreover, years of education is significantly negatively correlated with life satisfaction (*p* < 0.05). On the whole, there is no significant relationship between gender, registered residence, type of work, and life satisfaction (*p* > 0.05).

## 4. Discussion

In this study, we used the 2016 CFPS data to shed light on the associations between individual social capital and life satisfaction among mainland Chinese adults. We found that cognitive social capital has a stable positive impact on Chinese adults’ life satisfaction, whereas structural social capital has no such effect. These findings can provide reliable evidence for policy-making aimed at improving life satisfaction through individual social capital among adults in mainland China.

### 4.1. Cognitive Social Capital and Life Satisfaction

The higher the level of general trust, the higher the life satisfaction of adults, which is also confirmed by our study. A study using cross-country data found that high general trust leads to a high level of life satisfaction in most western Asian countries [29]. In the present study, the results from the binary logistic regression analysis confirm this, which is also consistent with other studies in Europe [38], the United States [39], and China [26]. Similar studies in rural China have shown that general trust exhibited strong and consistently positive associations with life satisfaction by facilitating social networks and emotional support [40]. Actually, the rapid development of China’s economy promotes social transformation. Subsequently, the crisis of trust is gradually spreading. As one of the fine traditional virtues of the Chinese nation, trust exists between people in an invisible form. Given that past studies have confirmed the strong role of general trust in promoting life satisfaction, there is an urgent need to restore general trust among the public.

As for “norms of reciprocity”, we found that reciprocity promotes the improvement of life satisfaction at the cognitive level, whereas unfair treatment inhibits life satisfaction in terms of historical experience. Such findings have been confirmed in other similar studies. Previous studies have found that reciprocity can indirectly reduce the incidence of depression by regulating social support [41], which may also promote the improvement of life satisfaction. A study in Boston pointed out that, regardless of the level of unfair treatment, it is related with higher rates of clinical depression among Puerto Ricans [42]. One study even found that, no matter the race, if middle-aged women experience daily unfair treatment, there is a significant impact on blood pressure [43]. Since ancient times, China has had a spirit of mutual assistance in which one side has difficulties and the other side supports them. However, with the rapid development of social economy, the income gap between all walks of life has widened and the problem of social equity has become increasingly prominent. Using these studies, the government needs to take effective measures to coordinate the relationship between different classes, maintain the relative fairness of society, help to form norms of reciprocity, and then improve the life satisfaction of adults.

Self-rated neighborhood relationship plays a significant positive role in life satisfaction. Keeping on good terms with neighbors can produce positive psychological states, foster health-related social norms, and ensure social support to cope with daily stress [44]. In addition, previous studies have pointed out that neighborhood relationship constitutes an important part of daily life [45]. As an old Chinese saying goes, a close neighbor is better than a distant relative, which fully expresses the status of friendly neighbor relationship in the hearts of the Chinese people. However, due to the complexity of social relationship and the decrease of adults’ sense of security in modern society, the neighborhood relationship, which was discussed almost everything in traditional Chinese society, has gradually become indifferent and strange. According to our findings, community departments should start to alter this situation and enhance the communication between neighbors, in order to better improve life satisfaction and effectively prevent the formation of adverse mental health diseases.

### 4.2. Structural Social Capital and Life Satisfaction

Organizational participation, however, has no significant relationship with life satisfaction after introducing sociodemographic variables and SES. As we mentioned earlier, previous studies on the impact of structural social capital on life satisfaction have produced quite confusing results. Similar conclusions have been drawn in other studies that organizational participation is not significantly associated with subjective well-being at the individual level [40]. The likely reason is that the enthusiasm for organizational participation was replaced by other community activities, which could better improve their enjoyment of life and maintain their mental health. Accordingly, considering the sociodemographic variables and SES, it seems reasonable that the impact of organizational participation on life satisfaction is no longer significant.

### 4.3. Socioeconomic Status and Life Satisfaction

Variables measuring SES show a positive effect on life satisfaction in mainland China. The higher the self-rated income level, the more people are satisfied with life. This is consistent with the findings of JeongHee Yeo and Yoon G. Lee [46], where they pointed out that, no matter the actual economic situation, self-rated income level substantially matters to life satisfaction. As expected, the effect of self-rated social status on life satisfaction is the same as that of self-rated income level. This is because when people acquire a higher social status, this is accompanied by the improvement of individual rights and the expansion of social networks. People become satisfied with their spiritual life and material life. Surprisingly, the years of education is inversely proportional to the level of life satisfaction in our study, which contradicts most previous studies [47,48]. The possible explanation is that the longer the years of education, the higher the demands on life, which leads to the decrease of life satisfaction.

### 4.4. Sociodemographic Variables and Life Satisfaction

The results of sociodemographic variables are in accordance with previous studies [49]. There is a similar U-shaped distribution between age and life satisfaction. It is reasonable for middle-aged people to feel dissatisfied with their lives under the double pressure from family and society. There is no difference in life satisfaction between men and women in our study, which is the same as previous studies [50]. Furthermore, people who are married or cohabited are more likely to be satisfied with their lives than others. In fact, similar studies in the past have reached the same conclusion. It has been suggested that marriage serves as an important protective shield against outside pressures in developing China [51]. On the whole, people who are married or cohabited can support each other to overcome difficulties and achieve higher life satisfaction.

## 5. Strengths and Limitations

The advantage of this study is that we selected national representative data to complete the study. The quality control team applied multiple methods of monitoring and intervention to ensure data quality. These practices laid the solid foundation for our study to obtain more reliable results. However, we must mention the following shortcomings. First, owing to the cross-sectional study, the causal relationship between individual social capital and life satisfaction needs to be confirmed by subsequent studies. Second, we limited social capital to personal resources in the process of study. It will be crucial for us to probe into the impact of contextual and individual social capital on life satisfaction in future studies. Lastly, we only selected a few single questions to measure each dimension of individual social capital. In future studies, enriching the diversity of indicators may better ensure the reliability of the study.

## 6. Conclusions

Through this study, we found that individual social capital can promote the improvement of life satisfaction to a certain extent. Given the stabilizing effect of cognitive social capital on life satisfaction at the individual level, the government should attach great importance to this aspect when trying to improve adults’ life satisfaction.

## Figures and Tables

**Table 1 ijerph-18-00441-t001:** Characteristics of the study sample.

Variables	Category	N	Weighted Composition Ratio (%(SE))
Age	16–25	3589	15.1 (0.3)
	26–35	3976	9.8 (0.2)
	36–45	4262	14.2 (0.3)
	46–55	5970	22.0 (0.3)
	56–65	4562	18.6 (0.3)
	>65	3650	20.3 (0.4)
Gender	male	1315	51.0 (0.4)
	female	1285	49.0 (0.4)
Registered residence	rural	1876	68.0 (0.4)
	urban	7245	32.0 (0.4)
Marital status	never married	4034	16.4 (0.3)
	married/cohabited	2018	75.0 (0.4)
	divorced/widowed	1794	8.6 (0.3)
Type of work	agriculture	9634	32.8 (0.4)
	non-agriculture	1078	37.2 (0.4)
	inapplicable	5587	30.0 (0.4)
Religious faith	no	2242	87.7 (0.3)
	yes	3589	12.3 (0.3)
Self-rated health	unhealthy	3866	15.1 (0.3)
	general	4647	19.1 (0.4)
	healthy	1749	65.8 (0.4)
Years of education	0–6	9739	38.7 (0.4)
	7–9	8519	31.5 (0.4)
	10–12	4483	19.4 (0.4)
	≥13	3268	10.4 (0.3)
Self-rated income level	low	1170	43.7 (0.4)
	general	9797	36.2 (0.4)
	high	2275	8.9 (0.3)
	inapplicable	2231	11.2 (0.3)
Self-rated social status	low	8569	31.6 (0.4)
	general	1204	45.9 (0.4)
	high	5391	22.5 (0.4)
General trust	trustworthy	1475	57.0 (0.4)
	suspicious	1125	43.0 (0.4)
Reciprocity	helpful	1875	73.0 (0.4)
	selfish	7259	27.0 (0.4)
Unfair treatment	never experienced	1904	73.4 (0.4)
	have experienced	6963	26.6 (0.4)
Neighborhood relationship	good	16,090	63.0 (0.4)
	poor	9919	37.0 (0.4)
Organizational participation	no	17,798	65.1 (0.4)
	yes	8211	34.9 (0.4)
Total		26,009	100

Weighted data are used. SE, standard error; SES, socioeconomic status.

**Table 2 ijerph-18-00441-t002:** Distribution of life satisfaction and univariate analysis.

Variables	Category	Life Satisfaction		χ^2^	*p*
		Dissatisfied (%(SE))	Satisfied (%(SE))		
Age	16–25	35.6 (1.1)	64.4 (1.1)	946.936	<0.001
	26–35	53.8 (1.1)	46.2 (1.1)		
	36–45	51.5 (1.0)	48.5 (1.0)		
	46–55	47.5 (0.8)	52.5 (0.8)		
	56–65	39.2 (1.0)	60.8 (1.0)		
	>65	26.5 (1.0)	73.5 (1.0)		
Gender	male	41.8 (0.6)	58.2 (0.6)	5.188	0.100
	female	40.4 (0.6)	59.6 (0.6)		
Registered residence	rural	42.2 (0.5)	57.8 (0.5)	28.267	<0.001
	urban	38.7 (0.8)	61.3 (0.8)		
Marital status	never married	38.9 (1.1)	61.1 (1.1)	33.359	0.001
	married/cohabited	42.1 (0.5)	57.9 (0.5)		
	divorced/widowed	36.8 (1.6)	63.2 (1.6)		
Type of work	agriculture	43.1 (0.7)	56.9 (0.7)	545.982	<0.001
	non-agriculture	47.8 (0.7)	52.2 (0.7)		
	inapplicable	30.6 (0.8)	69.4 (0.8)		
Religious faith	no	41.1 (0.5)	58.9 (0.5)	0.129	0.792
	yes	40.8 (1.2)	59.2 (1.2)		
Self-rated health	unhealthy	50.4 (1.2)	49.6 (1.2)	369.549	<0.001
	general	48.2 (1.0)	51.8 (1.0)		
	healthy	36.9 (0.5)	63.1 (0.5)		
Years of education	0–6	38.4 (0.7)	61.6 (0.7)	79.288	<0.001
	7–9	44.9 (0.7)	55.1 (0.7)		
	10–12	40.9 (1.0)	59.1 (1.0)		
	≥13	40.1 (1.2)	59.9 (1.2)		
Self-rated income level	low	54.4 (0.6)	45.6 (0.6)	1726.412	<0.001
	general	34.4 (0.7)	65.6 (0.7)		
	high	16.6 (1.1)	83.4 (1.1)		
	inapplicable	30.0 (1.3)	70.0 (1.3)		
Self-rated social status	low	58.4 (0.8)	41.6 (0.8)	2274.23	<0.001
	general	40.4 (0.6)	59.6 (0.6)		
	high	18.3 (0.7)	81.7 (0.7)		
General trust	suspicious	45.8 (0.7)	54.2 (0.7)	180.054	<0.001
	trustworthy	37.5 (0.6)	62.5 (0.6)		
Reciprocity	selfish	50.6 (0.8)	49.4 (0.8)	360.138	<0.001
	helpful	37.6 (0.5)	62.4 (0.5)		
Unfair treatment	never experienced	37.1 (0.5)	62.9 (0.5)	471.795	<0.001
	have experienced	52.1 (0.8)	47.9 (0.8)		
Neighborhood relationship	poor	50.7 (0.7)	49.3 (0.7)	588.388	<0.001
	good	35.4 (0.5)	64.6 (0.5)		
Organizational participation	no	43.6 (0.5)	56.4 (0.5)	123.371	<0.001
	yes	36.5 (0.7)	63.5 (0.7)		
Total		41.1 (0.4)	58.9 (0.4)		

Weighted data are used. SE, standard error; SES, socioeconomic status.

**Table 3 ijerph-18-00441-t003:** Binary logistic regression analyses for life satisfaction on individual social capital (0 = dissatisfied; 1 = satisfied).

Variables	Category	Model 1		Model 2		Model 3	
		OR (95% CI)	*p*	OR (95% CI)	*p*	OR (95% CI)	*p*
General trust	suspicious	1.000		1.000		1.000	
	trustworthy	1.111 (1.030–1.199)	0.007	1.118 (1.031–1.212)	0.007	1.088 (1.002–1.181)	0.045
Reciprocity	selfish	1.000		1.000		1.000	
	helpful	1.378 (1.267–1.498)	<0.001	1.316 (1.203–1.439)	<0.001	1.275 (1.164–1.396)	<0.001
Unfair treatment	never experienced	1.000		1.000		1.000	
	have experienced	0.609 (0.563–0.659)	<0.001	0.619 (0.568–0.674)	<0.001	0.684 (0.627–0.747)	<0.001
Neighborhood relationship	poor	1.000		1.000		1.000	
	good	1.703 (1.583–1.832)	<0.001	1.575 (1.457–1.703)	<0.001	1.554 (1.436–1.682)	<0.001
Organizational participation	no	1.000		1.000		1.000	
	yes	1.292 (1.197–1.395)	<0.001	1.246 (1.137–1.365)	<0.001	1.061 (0.966–1.165)	0.217
Years of education	0–6			1.000		1.000	
	7–9			0.780 (0.714–0.853)	<0.001	0.880 (0.799–0.970)	0.010
	10–12			0.787 (0.700–0.884)	<0.001	0.832 (0.728–0.950)	0.007
	≥13			0.685 (0.596–0.787)	<0.001	0.814 (0.689–0.962)	0.016
Self-rated income level	low			1.000		1.000	
	general			1.796 (1.648–1.958)	<0.001	1.831 (1.677–1.999)	<0.001
	high			3.147 (2.634–3.761)	<0.001	3.212 (2.675–3.858)	<0.001
	inapplicable			2.263 (1.956–2.619)	<0.001	1.671 (1.400–1.994)	<0.001
Self-rated social status	low			1.000		1.000	
	general			1.551 (1.421–1.693)	<0.001	1.526 (1.395–1.668)	<0.001
	high			4.088 (3.623–4.612)	<0.001	3.791 (3.352–4.289)	<0.001
Age	16–25					1.000	
	26–35					0.514 (0.423–0.624)	<0.001
	36–45					0.556 (0.449–0.688)	<0.001
	46–55					0.664 (0.538–0.819)	<0.001
	56–65					0.861 (0.690–1.074)	0.185
	>65					1.272 (1.000–1.619)	0.050
Gender	male					1.000	
	female					1.074 (0.993–1.162)	0.075
Registered residence	rural					1.000	
	urban					0.992 (0.897–1.097)	0.875
Marital status	never married					1.000	
	married/cohabited					1.364 (1.138–1.635)	0.001
	divorced/widows					1.198 (0.942–1.523)	0.141
Type of work	agriculture					1.000	
	non-agriculture					1.034 (0.937–1.140)	0.509
	inapplicable					1.507 (1.321–1.718)	<0.001
Self-rated health	unhealthy					1.000	
	general					1.137 (0.989–1.309)	0.072
	healthy					1.916 (1.695–2.166)	<0.001
Constants		0.809 (0.742–0.881)	<0.001	0.439 (0.391–0.493)	<0.001	0.247 (0.196–0.311)	<0.001

Weighted data are used. OR, odds ratio; CI, confidence interval; SES, socioeconomic status.

## Data Availability

The data can be obtained on: https://opendata.pku.edu.cn/dataverse/CFPS. Researchers are required to apply for permission to use the data.

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
