# Peer review of "Individual Social Capital and Life Satisfaction among Mainland Chinese Adults: Based on the 2016 China Family Panel Studies"

_ijerph, 2021, doi:10.3390/ijerph18020441_

Round 1
Reviewer 1 Report
The article deals with the relationship between individual social capital and life satisfaction among residents of mainland China. The aim of the research is clear and is reflected in the question asked in the title of the article. Although the issue of causality is indicated as a limitation to this study (that is, it is pointed that the cross-sectional data used in this study do not allow for examining if social capital influences life satisfaction; they only allow to investigate if a relationship between these two variables exists), the title of the article suggests a specific cause and effect relationship (i.e, that the direction of causality goes from social capital to satisfaction)e, which should be corrected.
Social capital is the focal variable of the research presented in the article. However, the paper lacks a theoretical framework that would serve as a conceptual foundation for the empirical examination demonstrated and discussed in the article. This is in line with a rather modest review of the literature in which the author(s) refer to prior empirical studies and not to theoretical considerations concerning social capital. For instance, the theory of social resources as presented and developed by Lin (1999, 2001) has not been addressed in the study. Similarly, the influential theory of social ties proposed by Granovetter (1973, 1979) has not been mentioned in the study. This is a serious weakness of the article which should be removed.
The consequence of the weakness mentioned above is a loose and discretionary selection of the variables representing the social capital in the study. It has not been explained why the authors chose these particular variables (that is, trust, reciprocity, neighborhood relationships, organizational participation), while omitting others.
The methodological section of the article clearly presents the data and methods used. The only reservation regarding this section refers to the abbreviated name of the computer-assisted personal interviewing which is normally shortened to CAPI and not CPI, as in the article. My doubts were also raised by the way the middle value on the Likert scale was marked (i.e., ‘general’). Nomen omen, this is very general and rather vague tag (I believe that in this particular study ‘Neither satisfied nor dissatisfied’ and ‘Neither good nor bad’ would be much more appropriate choice).
The section ‘Discussion’ hardly adds value. It resembles a literature review too much and helps to understand the results of the study too little. When reading this section I realized that the article does not situate this problem in the very intriguing context of the specifics of social life in China. Considering the characteristics of social life in China, it seems necessary to discuss the results of the study in this particular context. At the same time, this additional perspective in the section ‘Discussion’ could make it more interesting and less superficial.
Finally, the article needs a thorough proof editing as it suffers from many linguistic shortcomings.
Author Response
Dear Reviewer,
We would like to express our great thanks to you for giving us the opportunity to revise our manuscript entitled “Does individual-level social capital affect adult life satisfaction in China?” (ID: ijerph-1052177) and give us valuable advice. We have accepted all your suggestions and have tried my best to make some modifications.
The main corrections in the manuscript and the responds to the comments are as following:
Point 1: The article deals with the relationship between individual social capital and life satisfaction among residents of mainland China. The aim of the research is clear and is reflected in the question asked in the title of the article. Although the issue of causality is indicated as a limitation to this study (that is, it is pointed that the cross-sectional data used in this study do not allow for examining if social capital influences life satisfaction; they only allow to investigate if a relationship between these two variables exists), the title of the article suggests a specific cause and effect relationship (i.e, that the direction of causality goes from social capital to satisfaction), which should be corrected.
Response 1: We revised the title to: “Individual social capital and life satisfaction among mainland Chinese adults: based on the 2016 China Family Panel Studies”. Please see the change in line 2-4.
Point 2: Social capital is the focal variable of the research presented in the article. However, the paper lacks a theoretical framework that would serve as a conceptual foundation for the empirical examination demonstrated and discussed in the article. This is in line with a rather modest review of the literature in which the author(s) refer to prior empirical studies and not to theoretical considerations concerning social capital. For instance, the theory of social resources as presented and developed by Lin (1999, 2001) has not been addressed in the study. Similarly, the influential theory of social ties proposed by Granovetter (1973, 1979) has not been mentioned in the study. This is a serious weakness of the article which should be removed.
Response 2: It is really true as you suggested that the manuscript lacks a theoretical framework. In addition to Lin and Granovetter you mentioned, we find that Bourdieu, Coleman and Putnam have also done a lot of research in the field of social capital. Therefore, we further summarized relevant research in the hope of clarifying the theoretical framework of social capital. Please see the change in line 46-61.
Change 2: In the new version, we have added the following contents.
Line 46-61: Many scholars have their own unique views on social capital. Bourdieu defined social capital from the perspective of social networks and pointed out that social capital is a collection of actual or potential resources inseparable from the enduring network of relationships. On the basis of Bourdieu, Coleman defined social capital from its specific functions and pointed out that social capital is not a single entity, but different entities with various forms. Putnam noted that “social capital” refers to be “features of social organization”, in terms of trust, norms of reciprocity and social network. In addition to the above three different definitions of social capital, other authors also contribute to the enrichment and development of social capital. For example, Granovetter put forward the theory of "weak ties", in which individuals connect with other social circles to form Bridges that can provide valuable information that is not available in their own circles. Lin believed that social capital is a resource embedded in social networks and social relationship, which can be acquired or flowed through purposeful actions. To sum up, the definition of social capital has not been unified.
Point 3: The consequence of the weakness mentioned above is a loose and discretionary selection of the variables representing the social capital in the study. It has not been explained why the authors chose these particular variables (that is, trust, reciprocity, neighborhood relationships, organizational participation), while omitting others.
Response 3: This point is very excellent. As for the selection of social capital indicators, we mainly consider three aspects. Firstly, we have summarized the definitions and concepts of social capital according to previous studies. For example, Putnam has pointed out that social capital is related to trust, norms of reciprocity and social network in his understanding of social capital (Putnam, Robert D., Robert Leonardi, and Raffaella Y. Nanetti. "Making Democracy Work: Civic Traditions in Modem Italy." International Affairs 70, no. 1 (1993)). Secondly, we have drawn on from the previous studies of social capital measurement method. For example, Pan divided social capital into cognitive and structural in his research on the relationship between social capital and life satisfaction of the elderly in rural China. Cognitive social capital includes trust and family support, while structural social capital includes social membership and activity frequency (Pan H. Social Capital and Life Satisfaction across Older Rural Chinese Groups: Does Age Matter? Soc Work. 2018 Jan 1;63(1):75-84.). This kind of variable partition is somewhat similar to our research. Finally, according to the existing questionnaires in the database, we have chosen four variables (namely organizational participation, general trust, norms of reciprocity and neighborhood relationship) as the indicators to measure social capital at the individual level. In addition, we also realize that the manuscript has some limitations in the selection of social capital indicators, which are mentioned in the “Strengths and Limitations” of the article. Please see the change in line 69-73 and 321-323.
Change 3: In the new version, we have added the following contents.
Line 69-73: There is a growing consensus that empirical indicators of social capital can be measured by social networks (e.g. informal relationship, volunteering, organizational participation), social norms (e.g. shared norms, civic values) and trust (e.g. institutional trust, interpersonal trust, generalized trust). Similarly, some evidence can also be found to prove it in the above definition.
Line 321-323: Lastly, we have to only selected a few single questions to measure each dimension of individual social capital. In the future studies, enriching the diversity of indicators may better ensure the reliability of the study.
Point 4: The methodological section of the article clearly presents the data and methods used. The only reservation regarding this section refers to the abbreviated name of the computer-assisted personal interviewing which is normally shortened to CAPI and not CPI, as in the article. My doubts were also raised by the way the middle value on the Likert scale was marked (i.e., ‘general’). Nomenomen, this is very general and rather vague tag (I believe that in this particular study ‘Neither satisfied nor dissatisfied’ and ‘Neither good nor bad’ would be much more appropriate choice).
Response 4: The abbreviation name of the Computer Assisted Personal Interviewing was changed to CAPI. Please see the change in line 123. As for the dispute about the middle value of Likert, we change it from “general” to “3” (please see change in line 135) and “fair” (please see change in line 159). The reason for the modification of line 135 and 159 is that we reviewed the English version of the original questionnaire and find that it is expressed in these forms.
Change 4:
Line 123: Computer Assisted Personal Interviewing (CAPI)
Line 135: dissatisfied (answer from 1 to 3)
Line 159: poor (answers: “very poor”, “poor” and “fair”)
Point 5: “The section ‘Discussion’ hardly adds value. It resembles a literature review too much and helps to understand the results of the study too little. When reading this section I realized that the article does not situate this problem in the very intriguing context of the specifics of social life in China. Considering the characteristics of social life in China, it seems necessary to discuss the results of the study in this particular context. At the same time, this additional perspective in the section ‘Discussion’ could make it more interesting and less superficial.”
Response 5: Thanks very much for the comment. We made further discussion according to your suggestion combined with the characteristics of Chinese society. Please see the change in line 243-249, 258-263 and 270-276.
Change 5: In the new version, we have added the following contents.
Line 243-249: Actually, the rapid development of China’s economy promotes the social transformation. Then, the crisis of trust is gradually spreading. As one of the fine traditional virtues of the Chinese nation, trust exists between people in an invisible form. Given that past studies have confirmed the strong role of general trust in promoting life satisfaction, there is an urgent to restore general trust among the public.
Line 258-263: Since ancient time, China has had a spirit of mutual assistance in which one side has difficulties and the other side supports each other. However, with the rapid development of social economy, the income gap between all walks of life has widened and the problem of social equity has become increasingly prominent. By these studies, the government need to take effective measures to coordinate the relationship between different classes, maintain the relative fairness of society, help to form norms of reciprocity and then improve the life satisfaction of adults.
Line 270-276: However, due to the complexity of social relationship and the decrease of adults’ sense of security in modern society, the neighborhood relationship which has nothing to talk about in traditional Chinese society has gradually become indifferent and strange. According to our findings, the community departments should start to break the situation and enhance the communication between neighbors, so as to better improve life satisfaction and effectively prevent the formation of adverse mental health diseases.
Point 6: “The article needs a thorough proof editing as it suffers from many linguistic shortcomings.”
Response 6: According to your opinion, we checked the grammar in the manuscript and made some modifications. Please see the change in line 98-99,164-165,205,208, 230, 241 and 326. We are willing to use language services if we have enough time or it is necessary in the future process.
Change 6:
Line 98-99: Changed the verb “are” to “were”.
Line 164-165: If the respondent is a member of any of the above organizations, he will be classified as “yes”.
Line 205 and 208: Changed “lives” to “life”.
Line 230: We used the 2016 CFPS data to shed light on the associations between individual social capital and life satisfaction among mainland Chinese adults.
Line 241: Changed the verb “showed” to “shown”
Line 326: Through this study, we found that individual social capital can promote the improvement of life satisfaction to a certain extent.
Lastly, we deeply appreciate your consideration of our manuscript. I sincerely hope that our amendments can be consistent with your suggestions.
If it is very unfortunate that there are still deficiencies in our manuscript, we would appreciate and look forward to your valuable opinions again.
Sincerely,
Wenjun Li

Reviewer 2 Report
Despite the quality of the study and the large sample used, there is one aspect of the article that worries me, which is the validity of social capital measurement.
Before moving on to a more in-depth review, I would like to ask the authors to submit again the manuscript clarifying this aspect. Therefore, while there are numerous studies in the literature that report the validity of the measure of life satisfaction even with a single item (See for example: Cheung, Lucas, 2014, Assessing the Validity of Single-item Life Satisfaction Measures: Results from Three Large Samples, Quality of Life Research, 2014, DOI: 10.1007/s11136-014-0726-4), I do not find anything similar with respect to the way of measuring social capital through single item for each sub-dimension. The authors treat the 4 sub-dimensions of social capital as if there was a previous study that had demonstrated its psychometric characteristics, evaluating the reliability, the various forms of validity (f.e. construct and concurrent validity) and the factorial structure.
Presentlly, in my opinion the choices are not adequately justified (also the measure using subdivision into cognitive and structural aspects of social capital if was not validated in a previous study. I see the references, but it is not reported the psychometric features of the measure in the manuscript). I would ask the authors to provide the requested information so that I can review more in-depth, please.
Author Response
Dear Reviewer,
We would like to express our great thanks to you for giving us the opportunity to revise our manuscript entitled “Does individual-level social capital affect adult life satisfaction in China?” (ID: ijerph-152177) and give us valuable advice. We have accepted all your suggestions and have tried my best to make some modifications.
The main corrections in the manuscript and the responds to the comments are as following:
Point 1: Despite the quality of the study and the large sample used, there is one aspect of the article that worries me, which is the validity of social capital measurement. Before moving on to a more in-depth review, I would like to ask the authors to submit again the manuscript clarifying this aspect. Therefore, while there are numerous studies in the literature that report the validity of the measure of life satisfaction even with a single item (See for example: Cheung, Lucas, 2014, Assessing the Validity of Single-item Life Satisfaction Measures: Results from Three Large Samples, Quality of Life Research, 2014, DOI: 10.1007/s11136-014-0726-4), I do not find anything similar with respect to the way of measuring social capital through single item for each sub-dimension.
Response 1: Thanks for your affirmation of our research advantages. We would like to conclude that your first question is mainly about the validity of social capital measurement.
Firstly, the reason why we chose four variables as the measurement of social capital is that we have summarized the previous studies. These four variables can be found in the definition of social capital to support our choice. For example, Putnam has pointed out that social capital is related to trust, norms of reciprocity and social network in his understanding of social capital (Putnam, Robert D., Robert Leonardi, and Raffaella Y. Nanetti. "Making Democracy Work: Civic Traditions in Modem Italy." International Affairs 70, no. 1 (1993)). At the same time, we also realized that there was no description of this part in our manuscript. Therefore, we made corresponding changes in the manuscript according to your suggestion. Please see the changes in line 69-73. Moreover, we also realized that we have some defects in the selection of variables, so we pointed out it in the “Strengths and Limitations” section of the manuscript. Please see the changes in line 321-323.
Secondly, the data we used are from China Family Panel Studies (CFPS), funded by “985 program” of Peking University and the National Natural Science Foundation of China. CFPS is a national longitudinal general social survey project maintained by the Institute of Social Science Survey (ISSS) of Peking University. It adopted telephone check, field check, audio record check, interview reviews and statistical analysis to ensure data quality. Its data quality has also been widely recognized by scholars at home and abroad.
Lastly, we are very happy to provide you with several published papers, and the variables they used are single-item. (See for example: â‘ Han KM, Han C, Shin C, Jee HJ, An H, Yoon HK, Ko YH, Kim SH. Social capital, socioeconomic status, and depression in community-living elderly. J Psychiatr Res. 2018 Mar;98:133-140. â‘¡ Xue X, Cheng M. Social capital and health in China: exploring the mediating role of lifestyle. BMC Public Health. 2017 Nov 6;17(1):863.)
Change 1: In the new version, we have added the following contents.
Line 69-73: There is a growing consensus that empirical indicators of social capital can be measured by social networks (e.g. informal relationship, volunteering, organizational participation), social norms (e.g. shared norms, civic values) and trust (e.g. institutional trust, interpersonal trust, generalized trust). Similarly, some evidence can also be found to prove it in the above definition.
Line 321-323: Lastly, we only selected a few single questions to measure each dimension of individual social capital. In the future studies, enriching the diversity of indicators may better ensure the reliability of the study.
Point 2: The authors treat the 4 sub-dimensions of social capital as if there was a previous study that had demonstrated its psychometric characteristics, evaluating the reliability, the various forms of validity (f.e. construct and concurrent validity) and the factorial structure.
Presently, in my opinion the choices are not adequately justified (also the measure using subdivision into cognitive and structural aspects of social capital if was not validated in a previous study. I see the references, but it is not reported the psychometric features of the measure in the manuscript). I would ask the authors to provide the requested information so that I can review more in-depth, please.
Response 2: This point is very excellent. The division of structural and cognitive is common in the past research. Most of the literature we read adopted this method, combined with the variables we selected, we finally decided to use this method. For example, Pan divided social capital into cognitive and structural in his research on the relationship between social capital and life satisfaction of the elderly in rural China (Pan H. Social Capital and Life Satisfaction across Older Rural Chinese Groups: Does Age Matter? Soc Work. 2018 Jan 1;63(1):75-84.). Cao divided social capital into cognitive and structural two parts in the analysis of social capital and depression of urban elderly in China. There are still many studies using this method (Cao W, Li L, Zhou X, Zhou C. Social capital and depression: evidence from urban elderly in China. Aging Ment Health. 2015;19(5):418-29.). Please see the changes in line 62-63. Moreover, we are not sure whether our understanding of the question is accurate, and if the answer is not consistent with your suggestion, please comment again.
Change 2:
Line 62-63: Moreover, social capital is often used to examine its positive effects on mental health in terms of its structural or cognitive components, such as depression and life satisfaction.
Lastly, we deeply appreciate your consideration of our manuscript. I sincerely hope that our amendments can be consistent with your suggestions.
If it is very unfortunate that there are still deficiencies in our manuscript, we would appreciate and look forward to your valuable opinions again.
Sincerely,
Wenjun Li

Round 2
Reviewer 1 Report
Thank you for the responses and revisions made in the manuscript. In my opinion the paper still needs a professional proof reading.
Reviewer 2 Report
I thank the authors for the responses. I was already underlining the quality of the work done and my doubts mainly concerned on theoretical aspects in support of the empirical choices that authors clarified to me. I consider of crucial importance their inclusion to the social capital literature extension in the introduction, also it was really appropriate to have reported the new sentence in the "Strengths and Limitations" section as a limit to the measurement of social capital. By analyzing the manuscript with the parts added, now I think that the meaning of the work done is clearer. I have nothing else to report, so I suggest to accept the article in this form.